# OmniDFA: A Unified Framework for Open Set Synthesis Image Detection and Few-Shot Attribution

## Abstract

AI-generated image (AIGI) detection and source model attribution remain central challenges in combating deepfake abuses, primarily due to the structural diversity of generative models. Current detection methods are prone to overfitting specific forgery traits, whereas source attribution offers a robust alternative through fine-grained feature discrimination. However, synthetic image attribution remains constrained by the scarcity of large-scale, well-categorized synthetic datasets, limiting its practicality and compatibility with detection systems. In this work, we propose a new paradigm for image attribution called open-set, few-shot source identification. This paradigm is designed to reliably identify unseen generators using only limited samples, making it highly suitable for real-world application. To this end, we introduce OmniDFA (Omni Detector and Few-shot Attributor), a novel framework for AIGI that not only assesses the authenticity of images, but also determines the synthesis origins in a few-shot manner. To facilitate this work, we construct OmniFake, a large class-aware synthetic image dataset that curates $1.17$ M images from $45$ distinct generative models, substantially enriching the foundational resources for research on both AIGI detection and attribution. Experiments demonstrate that OmniDFA exhibits excellent capability in open-set attribution and achieves state-of-the-art generalization performance on AIGI detection. The integration of the new task enhances detection performance and offers an efficient and scalable path toward practical adoption.

## 1 Introduction

Generative models now forge photorealistic images that defy visual scrutiny, collapsing the boundary between authentic and synthetic. The growing threat of widespread misuse requires operational methods to not only distinguish AI-generated images (AIGI) from real ones but also trace their generative origins, which is critical for understanding model-specific vulnerabilities. However, existing tools for detecting and analyzing synthetic content struggle to match the pace of advancing generation methods. This poses a challenge to such countermeasures in an open-set scenario, where they must handle data from generative models unseen during training, necessitating robust generalization ability and strong fine-grained discrimination capability.

Current detection methods aim to extract universal artifacts as discriminative features, yet a common practice (Zhu et al., 2023) of training on data from a single generator leads to overfitting on model-specific biases and ignores feature diversity, ultimately impairing the generalization ability. The coarse-grained nature of the binary classification task fundamentally limits its ability to capture the rich variety of features. In contrast, image attribution presents itself as a broader and more fine-grained task for forgery analysis. However, it remains in its early stages and far from settled, primarily due to two limitations. First, large-scale, well-categorized attribution datasets are still scarce. Existing synthetic image datasets (Wang et al., 2020; Bird & Lotfi, 2024) either originate from a relatively small number of generators, or categorize images based on non-architectural characteristics, which are unsuitable for model attribution. Second, current image attribution paradigms lack practicality. The two commonly used tasks each have significant shortcomings: closed-set identification can only trace images back to generators seen during the training phase, while open-set rejection simply categorizes all samples from unknown sources into a single unknown class, which

| In Set (Known Generators) | Out of Set (Unknown Generators) | Real | Support Samples |

**(a) Authenticity Detection** · **(b) Closed-Set Identification** · **(c) Open-Set Rejection** · **(d) Open-Set Identification**

Figure 1: Comparison of task-specific pipelines for synthetic analysis. Our new attribution paradigm (d) offers dramatically improved scalability over previous works (b) and (c).

impedes further investigation into the distinctions among them. Overall, existing AIGI attribution methods are constrained to identifying only known generator types. Adapting these approaches to incorporate new categories necessitates retraining the entire network, which is resource-intensive and consequently makes them impractical for real-world deployment.

In this work, we investigate image attribution from a novel perspective by identifying the sources of generated images in an open-set scenario. We observe that generative models produce images with pronounced and model-specific biases, which can be effectively learned from a minimal number of support samples. To leverage this insight, we introduce the task of open-set few-shot identification, as depicted in Figure 1 (d). This task establishes a new paradigm for image attribution, which challenges models to identify the specific biases of unknown generators using only a few reference samples. We argue that this out-of-distribution identification task is more reliable and robust for evaluating model generalization, precisely because it demands the model to quantify, rather than merely identify, model-specific biases. The few-shot-based setting makes this new paradigm highly applicable and scalable. Its traceability for new generative models requires only a minimal number of samples, which significantly reduces the operational costs.

To facilitate our work, we begin by addressing the lack of a suitable dataset. To this end, we propose OmniFake, a large-scale, class-aware dataset specifically designed for multi-class attribution. We collect generated images from 45 distinct generative models spanning GANs, diffusion models, autoregressive models, and hybrid architectures. We maintain a substantial collection of synthetic images for each class, ensuring comprehensive coverage and diversity. OmniFake significantly surpasses prior datasets (Zhu et al., 2023; Zhong et al., 2024) in both model diversity and up-to-date coverage, with a clear comparison presented in Table 1. The defining feature of OmniFake is its strict enforcement of model heterogeneity. In

Table 1: Comparison with existing AIGI datasets. "Distinct" indicates that each generator in the dataset has a distinct architecture.

| AIGI Dataset | Generators | Fake Images | Distinct |
|---|---|---|---|
| CNNSpot | 11 | 362 K | ✓ |
| DiffusionForensics | 11 | 439 K | ✓ |
| GenImage | 8 | 1.33 M | ✗ |
| UniversalFakeDetect | 19 | 400 K | ✗ |
| Artifact | 25 | 2.49 M | ✗ |
| AIGCBenchmark | 17 | 360 K | ✗ |
| ImagiNet | 8 | 100 K | ✓ |
| WildFake | 23 | 2.55 M | ✗ |
| DRCT-2M | 16 | 2.00 M | ✗ |
| OmniFake | 45 | 1.17 M | ✓ |

contrast to previous datasets, where differences in weights, sampling steps, or adaptation modules often resulted in highly similar forgery clues, our dataset effectively supports the critical task of image attribution. This significant advancement substantially enriches the fundamental resources available for both AIGI detection and attribution research.

Leveraging our comprehensive dataset, we propose OmniDFA (Omni Detector and Few-shot Attributor), an innovative framework that simultaneously addresses authenticity detection and open-set few-shot identification. We adopt a dual-path architecture that captures image characteristics from both low-level and high-level perspectives to effectively capture fine-grained details while maintaining global representations. We employ supervised contrastive learning (Khosla et al., 2020) to effectively isolate and quantify model-specific biases. This approach encourages the learning of a more comprehensive forgery subspace that incorporates shared fake features, thereby improving generalization capability. To constrain the position of authentic data within the feature space, we specifically process the real images by employing center loss (Wen et al., 2016) to establish a compact centroid and learning a decision boundary that optimally encloses the data.

Figure 2: Samples of the OmniFake dataset. Our dataset covers a broad spectrum of generative models, with real images sourced from multiple datasets for comprehensive coverage.

Experimental results show that OmniDFA achieves state-of-the-art performance on our proposed OmniFake dataset, surpassing previous method by 5.83% in authenticity detection accuracy. Additionally, it exhibits robust zero-shot detection performance across various benchmarks, confirming the generalizability and effectiveness of our approach.

Our contributions are summarized as follows:

- We establish open-set few-shot identification as a previously unexplored task for evaluating model generalization, which better matches practical constraints.

- We construct the OmniFake dataset, a large-scale, class-aware synthetic image dataset that enables in-depth study of model-specific patterns for image attribution.

- We propose OmniDFA, a novel AIGI analyzer that integrates authenticity detection with open-set source identification in a cohesive framework.

- Extensive experiments demonstrate the effectiveness of our OmniDFA, which significantly outperforms current state-of-the-art methods in both detection and attribution tasks.

## 2 CONSTRUCTION OF OMNIFAKE

As provided in Table 1, most million-scale datasets do not emphasize the uniqueness of each category, collecting data from generators with similar model architectures. To support our investigation in multi-class attribution, we carefully construct a dataset comprising fake images sampled from a diverse set of models. For a visual overview, a snapshot of our dataset is shown in Figure 2.

### 2.1 FAKE IMAGE COLLECTION

To ensure comprehensive data diversity, we gather fake images through three distinct channels: (1) established datasets and benchmarks, (2) community-shared collections, and (3) synthetic images generated using open-source models. We curate data from open-source datasets including GenImage (Zhu et al., 2023), WildFake (Hong et al., 2025), and MPBench (Lu et al., 2023). The final collection of this channel comprises images produced by 19 distinct generators spanning GANs, diffusion models, and VAEs. To enrich our dataset with data from closed-source models, we selected 8 synthetic dataset on Hugging Face (HuggingFace, 2016), such as DALLE3 (OpenAI, 2023), Ideogram (IdeogramAI, 2023), and Midjourney V6 (Midjourney, 2021). We also select 18 state-of-the-art open-source models from Hugging Face or their official repositories, generating corresponding images based on a comprehensive collection of prompts. These include flow-matching models such as Hunyuan-DiT (Li et al., 2024) and SD3-Medium (Esser et al., 2024), as well as autoregressive models like Janus-pro (Chen et al., 2025b), BAGEL (Deng et al., 2025), Show-o (Xie et al., 2025). Additionally, we incorporate several cutting-edge multimodal unified models that utilize diffusion-based decoders for high-fidelity image generation, including OmniGen2 (Wu et al., 2025a), Ovis-U1 (Wang et al., 2025a), and UniWorld-V1 (Lin et al., 2025), among others. Details on the image generators and the synthesis pipeline are provided in Appendix C.

Figure 3: Overview of OmniDFA. Our dual-path architecture captures both low-level and high-level image features, balancing fine details and global representations. We use contrastive learning to enhance feature discrimination and apply sphere center loss to compactly cluster real samples.

## 2.2 REAL IMAGE COLLECTION

To establish a comprehensive benchmark for image forensics research, we curate a diverse collection of authentic images from 10 publicly available datasets spanning multiple domains. We carefully curate our dataset from two distinct categories: (1) Large-scale raw datasets including LAION (Schuhmann et al., 2021), WuKong (Gu et al., 2022), and CC12M (Changpinyo et al., 2021), providing vast amounts of in-the-wild data; (2) Carefully constructed datasets such as ImageNet (Russakovsky et al., 2015) and COCO (Lin et al., 2014), offering high-quality images from specific domains. This approach ensures a broad and representative set of real images, facilitating robust evaluation of forensic techniques across varied contexts.

## 2.3 ANALYSES OF OMNIFAKE

The OmniFake dataset comprises a total of 2.34 M images in the training set, with a balanced distribution of 1.17 M real images and 1.17 M synthetic images. The synthetic images originate from 45 distinct generative models that span a broad spectrum of architectures. Sample images from our dataset are illustrated in Figure 2. To ensure sufficient intra-class diversity, we guarantee that each synthetic image category in the training set contains at least 20 K samples. We also construct a separate test set comprising 90 K synthetic images, with 2 K samples per category. We include an equal number of real images, resulting in a balanced test set of 180 K images in total. OmniFake comprises images from cutting-edge generative models and features structural heterogeneity across different categories, making it the most extensive dataset for the attribution task. It provides a rich foundation for investigating the impact of model architectures on generalization, which may contribute to the advances in both AIGI detection and attribution research.

## 3 METHOD

In this section, we present our OmniDFA in detail, which is a novel AI-generated image (AIGI) framework that jointly addresses authenticity detection and few-shot open-set identification. Figure 3 illustrates the overall architecture of our proposed method.

## 3.1 MULTI-LEVEL FEATURE EXTRACTOR

Since current generative models are able to produce high-resolution images, resizing inputs to a fixed scale as in prior studies (Zhu et al., 2023) tends to degrade fine-grained details. Conversely, maintaining the original resolution without resizing often results in information loss due to computational constraints or limitations of model input. To address this problem, we propose a dual sampling strategy that captures both local and global features, thereby maximizing the retention of critical visual information across varying image resolutions.

As illustrated in Figure 3, OmniDFA processes a given image $\mathbf{x}_i$ through two complementary pathways: (1) an aspect-ratio-preserving resize, where the shorter edge is scaled to the target input size followed by center cropping, to retain holistic global features; and (2) direct high-resolution crop from the original image, which maximally preserves fine-grained textures and local details. The

distinct global and local features from the high-level and low-level branches are channel-wise concatenated and subsequently processed through a multi-layer perceptron (MLP) to produce the final feature representation $f_\theta(\mathbf{x}_i)$, where $\theta$ denotes the set of all trainable parameters in the network.

## 3.2 SUPERVISED CONTRASTIVE LOSS FOR ATTRIBUTION

To learn a highly discriminative and expressive feature representation for image attribution, we employ supervised contrastive learning and subdivide fake samples into fine-grained categories based on their generative sources. Meanwhile, all authentic samples remain grouped under a single class and are jointly trained with fake ones. This separation guides the model to distribute features of different forgery types across distinct regions in the embedding space. Thus, unseen fake samples are likely to fall between these regions, as they share partial artifacts with known forgery types.

Supervised contrastive learning optimizes feature embedding spaces through simultaneously minimizing intra-class variations while maximizing inter-class separability. Specifically, for a batch of $N$ samples $\{\mathbf{x}_1, \mathbf{x}_2, \ldots, \mathbf{x}_N\}$ with known categories, we first extract their features $f_\theta(\mathbf{x}_i)$ and then apply L2-normalization to obtain the corresponding vectors $\mathbf{z}_i = f_\theta(\mathbf{x}_i)/||f_\theta(\mathbf{x}_i)||_2$. The supervised contrastive loss can be formulated as follows:

$$\mathcal{L}_{sup} = \sum_{i=1}^{N} \frac{-1}{|P(i)|} \sum_{p \in P(i)} \log \frac{\exp(\mathbf{z}_i \cdot \mathbf{z}_p/\tau)}{\sum_{a \in A(i)} \exp(\mathbf{z}_i \cdot \mathbf{z}_a/\tau)}, \tag{1}$$

where $P(i)$ denotes the set of samples in the same class with $\mathbf{x}_i$, $A(i)$ represents all other samples in $P(i)$ except $p$, and $\tau$ denotes the temperature factor that scales the similarity scores.

## 3.3 SPHERE CENTER LOSS WITH LEARNABLE BOUNDARY

Considering the substantial quantity of real images may lead to feature dispersion in the embedding space, we implement a sphere center loss specifically for the real category to better regularize its feature distribution and enhance clustering. To ensure compatibility with the feature space of contrastive training, we directly apply the center loss on the normalized vector $\mathbf{z}_i$, formulated as:

$$\mathcal{L}_{cen} = \frac{1}{|P_r|} \sum_{p \in P_r} (1.0 - \mathbf{z}_p \cdot \mathbf{c}_r), \tag{2}$$

where $\mathbf{c}_r$ is a learnable L2-normalized vector representing the feature center of the real class, and $P_r$ denotes the set of all real samples in the current mini-batch. By unifying these constraints, we formulate the final learning objective with the scaling factor $\lambda$:

$$\mathcal{L} = \mathcal{L}_{sup} + \lambda \mathcal{L}_{cen}. \tag{3}$$

To enable direct measurement of image authenticity, we introduce a boundary threshold $\gamma$ defined as the maximum angular separation between real image features and the real center. We compute the higher bound using Tukey's fences and update it via momentum-based adjustment:

$$\gamma^- \leftarrow \beta\gamma^- + (1 - \beta)(Q_3 + 1.5(Q_3 - Q1)), \tag{4}$$

where $Q_1$ and $Q_3$ are the first and third quartiles of the deviation distribution, and $\beta$ is the momentum coefficient controlling the update strength. This update does not involve backpropagation but is performed numerically based on real sample inputs. During detection, samples exhibiting angular separation beyond $\gamma$ are classified as synthetic.

## 4 EXPERIMENT

In this section, we first present involved benchmarks and our experimental configurations. We then conduct comprehensive comparisons between OmniDFA and several state-of-the-art synthetic image detection and attribution models, evaluating performance on both authenticity discrimination and open-set fake category identification. We primarily focus on evaluating the zero-shot capability of models, examining their performance across unseen categories.

Table 2: Results of open-set few-shot identification on OmniFake. Each task is evaluated with 5 support samples. We report accuracy in percentage, with the best results highlighted in boldface.

| Open-Set Few-Shot Classifiers | OmniFake Dataset | | | | | | Averages (%) | |
| | Part I | | Part II | | Part III | | | |
| | 5-way | 15-way | 5-way | 15-way | 5-way | 15-way | 5-way | 15-way |
|---|---|---|---|---|---|---|---|---|
| DNA-Det (Yang et al., 2022) | 43.93 | 21.85 | 44.40 | 23.36 | 46.13 | 25.64 | 45.15 | 23.62 |
| CPL (Sun et al., 2023) | 42.18 | 23.22 | 43.88 | 24.47 | 51.17 | 34.00 | 45.74 | 29.46 |
| SiameseNet (Abady et al., 2024) | 41.64 | 27.66 | 43.88 | 28.86 | 46.42 | 31.85 | 43.98 | 29.46 |
| UniversalAttr (Cioni et al., 2024) | 48.85 | 29.17 | 52.11 | 30.35 | 56.27 | 38.41 | 52.41 | 32.64 |
| ComFor (Park & Owens, 2025) | 53.90 | 33.09 | 56.03 | 33.92 | 56.47 | 36.68 | 55.47 | 34.56 |
| FSD (Wu et al., 2025b) | 63.96 | 40.16 | 73.09 | 50.85 | 76.38 | **58.12** | 71.14 | 49.71 |
| OmniDFA | **65.88** | **40.53** | **75.33** | **52.52** | **77.37** | 57.66 | **72.86** | **50.24** |

## 4.1 BENCHMARKS AND EVALUATION METRICS

**Datasets and benchmarks.** We first conduct comprehensive experiments on our OmniFake dataset. To adapt our framework for the zero-shot task, we perform our experiments using 3-fold cross-validation by randomly dividing the OmniFake dataset into three balanced parts. In each validation round, we perform training on two parts while using the remaining part for testing. For the open-set few-shot attribution task, the evaluation is conducted under both 5-way 5-shot and 15-way 5-shot scenarios to assess model capability in recognizing novel categories.

To comprehensively evaluate the generalization capability of our model, we conduct extensive zero-shot experiments on the GenImage dataset (Zhu et al., 2023), comprising images generated by 8 generative models. Furthermore, we also perform additional comparative experiments on the Chameleon dataset (Yan et al., 2025). This challenging benchmark contains over 11 K high-fidelity AI-generated images collected from diverse online sources, specifically designed to simulate real-world application scenarios.

**Evaluation metrics.** Following established practices in previous research (Zhu et al., 2023; Wu et al., 2025b), we employ accuracy (ACC) and average precision (AP) as our evaluation metrics, with the threshold step for AP computation set to 0.05.

## 4.2 EXPERIMENTAL SETTINGS

We adopt the ConvNeXt-Small Liu et al. (2022) pretrained on ImageNet as the feature extractor of our model, which outputs a vector of 512 dimensions. These features are then processed through an MLP to produce the final 128-dimensional embeddings. Following Park & Owens (2025), we employ RandAugment (Cubuk et al., 2020), Gaussian blur, and JPEG compression during classifier training to effectively mitigate potential biases while enhancing the robustness of model.

During training, we sample fake images with a per-GPU batch size of 128 and real images with a per-GPU batch size of 16, resulting in a total batch size of 1152 to meet the requirements of contrastive learning. We use AdamW as our optimizer with a base learning rate of $2e^{-5}$ and employ a CosineAnnealing learning rate scheduler for a total of 20 epochs. We set the temperature parameter $\tau = 0.07$, loss coefficient $\lambda = 0.01$, and momentum update parameter $\beta = 0.99$. Additional implementation details can be found in Appendix E.2. Our method is implemented with the PyTorch library and all the experiments are conducted on 8 A100 with 40 GB memory.

## 4.3 OPEN-SET FEW-SHOT IDENTIFICATION

The out-of-distribution classification task serves as a crucial testbed for evaluating whether a model genuinely learns essential features of unseen categories, rather than simply memorizing training patterns. As shown in Table 2, our study evaluates a selection of existing methods, including four attribution approaches and two synthetic image detection techniques. The attribution methods are DNA-Det (Yang et al., 2022), CPL (Sun et al., 2023), SiameseNet (Abady et al., 2024), and UniversalAttr (Cioni et al., 2024). We also include AIGI detection methods FSD (Wu et al., 2025b) and ComFor (Park & Owens, 2025), motivated by their extensive training across a large number of

Table 3: Results of authenticity detection on OmniFake. We evaluate zero-shot detection performance across different dataset partitions, reporting both real and fake accuracy (Acc) and average precision (AP) in percentage. The best results are highlighted in boldface.

| Authenticity Detection Methods | OmniFake Dataset | | | | | | | | | | | | Averages (%) | |
|---|---|---|---|---|---|---|---|---|---|---|---|---|---|---|
| | Part I | | | | Part II | | | | Part III | | | | | |
| | F-Acc | R-Acc | Acc | AP | F-Acc | R-Acc | Acc | AP | F-Acc | R-Acc | Acc | AP | Acc | AP |
| UnivFD (Ojha et al., 2023b) | 7.39 | 98.66 | 53.03 | 56.07 | 24.08 | 98.22 | 61.15 | 67.78 | 16.43 | 98.64 | 57.54 | 62.14 | 57.24 | 62.00 |
| NPR (Tan et al., 2024a) | 74.13 | 82.18 | 78.16 | 76.98 | 75.99 | 82.21 | 79.10 | 78.38 | 79.26 | 81.93 | 80.60 | 80.35 | 79.28 | 78.57 |
| AIDE (Yan et al., 2025) | 77.51 | 96.28 | 86.90 | 94.01 | 83.20 | 96.32 | 89.76 | 94.62 | 78.53 | 96.20 | 87.37 | 93.68 | 88.01 | 94.10 |
| SAFE (Li et al., 2025a) | 76.93 | 97.76 | 87.35 | 91.46 | 73.73 | 97.61 | 85.67 | 92.01 | 65.06 | 97.67 | 81.37 | 88.94 | 84.79 | 90.80 |
| ComFor (Park & Owens, 2025) | 75.58 | **98.96** | 87.27 | 89.78 | 80.78 | **99.06** | 89.92 | 92.50 | 85.06 | **99.07** | 92.07 | 97.10 | 89.75 | 93.13 |
| FSD (Wu et al., 2025b) | 90.66 | 77.15 | 83.91 | - | 83.33 | 77.80 | 80.57 | - | 90.51 | 75.63 | 83.07 | - | 82.51 | - |
| OmniDFA | **97.43** | 95.32 | **96.38** | **97.56** | **96.36** | 93.63 | **95.00** | **95.93** | **97.06** | 93.65 | **95.36** | **97.42** | **95.58** | **96.97** |

categories, which suggests a strong potential for capturing diverse feature distributions. Although these models are not specifically designed for this task, we extract features from the last layer of their backbone and employ prototypical classification. Specifically, we compute the prototype centers from the support set and classify each query sample based on its distance to these centers. We evaluated each model on OmniFake using both 5-way 5-shot and 15-way 5-shot settings across 15 unseen categories from each part. To ensure statistical reliability, we conducted 10,000 independent test episodes for each experimental configuration following the episodic testing protocol.

As shown in Table 2, our model exhibits strong few-shot learning capabilities, confirming its effectiveness in identifying spurious category features and facilitating subsequent research. FSD also demonstrates competitive performance, underscoring the effectiveness of metric learning for this task. To our surprise, the standard binary classifier ComFor, though untrained for multi-class tasks, inherently exhibits certain classification abilities. This can be partly attributed to category overlap in its training data, as well as its powerful feature extraction capacity developed through large-scale multi-category training. Other attribution methods perform inadequately in open-set few-shot identification, primarily for two reasons: first, they often treat unseen categories as a single class without fine-grained distinction; second, their training sets cover limited categories, leading to overfitting on a small number of classes. Our results significantly surpass previous methods, demonstrating the effectiveness of our framework and highlighting promising directions for future research.

## 4.4 Cross-Generator Authenticity Detection

Our authenticity detection evaluation prioritizes the zero-shot capability of classifiers, defined as the generalization performance over previously unseen categories. Our experiments use the OmniFake dataset and compare against state-of-the-art methods, including UnivFD (Ojha et al., 2023b), NPR (Tan et al., 2024a), AIDE (Yan et al., 2025), SAFE (Li et al., 2025a), ComFor (Park & Owens, 2025), and FSD (Wu et al., 2025b). While these approaches were trained on their respective datasets, we emphasize their adaptability to novel data, as the ultimate goal is to achieve robust generalization beyond the training environment. Accordingly, we use their officially released weights to ensure optimal generalization. For thorough benchmarking against FSD based on metric learning, we retrain this model on our dataset. We include ComFor in our comparisons to benchmark against large-scale pretrained models. Although it incorporates a series of models built upon Stable Diffusion, potentially introducing dataset overlap, we consider this risk tolerable given the vast number of our test categories. The results are summarized in Table 3.

We observe that our OmniDFA consistently outperforms prior works in fake detection performance across all components of OminFake, achieving an average improvement of +5.83%. This demonstrates the strong generalization capability of our model. We also notice that our model exhibits slightly lower accuracy in real image detection. This is attributed to our boundary update rule, which actively filters out anomalous data and may consequently exclude some marginal cases. Additionally, ComFor achieves state-of-the-art performance in real image detection, which may be attributed to potential dataset overlap between its training data and our test set.

Furthermore, our results indicate that training with multiple generators yields better performance compared to single-generator training approaches like UnivFD, which shows limited effectiveness when applied to out-of-distribution datasets. Another observation is that FSD demonstrates good

Table 4: The zero-shot detection results on GenImage and Chameleon datasets. The best results are highlighted in boldface, while the second-optimal results are marked with underline. Our method demonstrates remarkable improvements in detection accuracy.

| Methods | GenImage | | | | | | | | | Chameleon | | |
|---|---|---|---|---|---|---|---|---|---|---|---|---|
| | ADM | Glide | Midjourney | SD v1.4 | SD v1.5 | VQDM | wukong | BigGAN | Average | F-Acc | R-Acc | Acc |
| CNNSpot | 60.39 | 58.07 | 51.39 | 50.57 | 50.53 | 56.46 | 51.03 | 71.17 | 56.20 | 9.86 | 99.55 | 60.89 |
| UnivFD | 66.87 | 62.46 | 56.13 | 63.66 | 63.49 | 85.31 | 70.93 | 95.08 | 70.49 | **85.52** | 41.56 | 60.42 |
| DIRE | 75.78 | 71.75 | 58.01 | 49.74 | 49.83 | 53.68 | 54.46 | 70.12 | 60.42 | 2.09 | 99.73 | 57.83 |
| PatchCraft | 82.17 | 83.79 | 90.12 | 95.38 | 95.30 | 88.91 | 91.07 | 95.80 | 90.32 | 1.39 | 96.52 | 55.70 |
| NPR | 69.69 | 78.36 | 77.85 | 78.63 | 78.89 | 78.13 | 76.61 | 84.35 | 77.81 | 1.68 | **100.00** | 57.81 |
| AIDE | **93.43** | 95.09 | 77.20 | 93.00 | 92.85 | 95.16 | 93.55 | 83.95 | 90.53 | 26.8 | 95.06 | 65.77 |
| OmniDFA | 85.50 | **96.71** | **97.58** | **97.47** | **97.75** | **96.78** | **97.78** | **97.33** | **95.86** | 77.46 | 88.00 | **83.48** |

performance in fake detection but performs poorly on real datasets. This limitation stems from its classification mechanism that relies on the nearest prototypical centroid, which becomes less effective when dealing with diverse types of fake images. In contrast, our model explicitly addresses the distribution of real images by incorporating a center loss term, which pulls the features of real samples closer to their class center in the embedding space, thereby enhancing the discriminative power for authentic data and significantly boosting the overall detection accuracy.

## 4.5 CROSS-DATASET EVALUATION

To comprehensively assess the generalization capability of OmniDFA, we selected two representative datasets: GenImage and Chemeleon, which serve as benchmarks for standard experimental evaluation and real-world application scenarios, respectively. To rigorously ensure experimental reliability, we train an additional classifier by explicitly excluding all categories in both benchmarks and their related model families. For instance, we completely removed SD1.5, SDXL, and SD3-Medium generated samples from our training data to ensure strictly zero-shot testing condition. We incorporate three additional baseline methods: CNNSpot Wang et al. (2020), DIRE (Wang et al., 2023a), and PatchCraft (Zhong et al., 2024). All comparative models are developed following Yan et al. (2025). Since the Chemeleon dataset does not provide fine-grained image categories, we only report the performance on real and fake images. Table 4 presents the comparative results between our OmniDFA and other methods.

As shown in Table 4, OmniDFA achieves remarkably high performancein these challenging zero-shot scenarios. Our model achieves state-of-the-art classification accuracy on the GenImage benchmark, further validating the effectiveness of our multi-generator training strategy. Chemeleon serves as a high-quality benchmark for evaluating real-world application scenarios. Existing models show limited fake image detection capability on this benchmark, indicating they overfit to authentic images in datasets like GenImage and fail to learn generalizable deepfake detection features. Our model outperforms the second-best approach by +17.71% in detection accuracy, while maintaining balanced performance across both image types, which demonstrates its strong practical applicability.

## 4.6 ABLATION STUDY

**Impact of compression and blurring.** Since most real-world datasets use JPEG format while synthetic data typically employs PNG format, we conducted JPEG compression and Gaussian blurring experiments to examine how image quality affects model performance. Using OmniFake part I as our test dataset, we evaluate the impact of JPEG compression on fake images and Gaussian noise on both images, as shown in Figure 4. The vertical dotted line indicates the augmentation boundaries adopted during our training.

The results demonstrate that our model maintains robust performance within the augmentation range but experiences noticeable degradation beyond these boundaries. In contrast, other methods show significant performance drops even from the outset. Although ComFor demonstrates excellent performance in handling blur due to corresponding enhancements, it still underperforms our method in compression scenario. These findings confirm that image quality substantially impacts detection performance, while proper augmentation techniques can effectively mitigate this effect. Additionally,

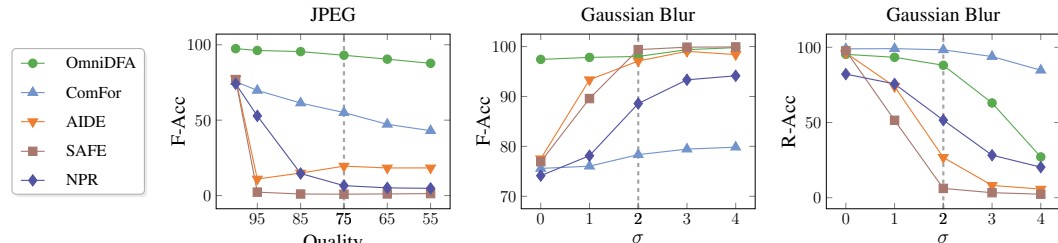

Figure 4: Robustness to compression and blurring. The vertical lines indicate the perturbation bounds used during our training. Most studies exhibit great sensitivity to these degradation factors when trained without perturbations.

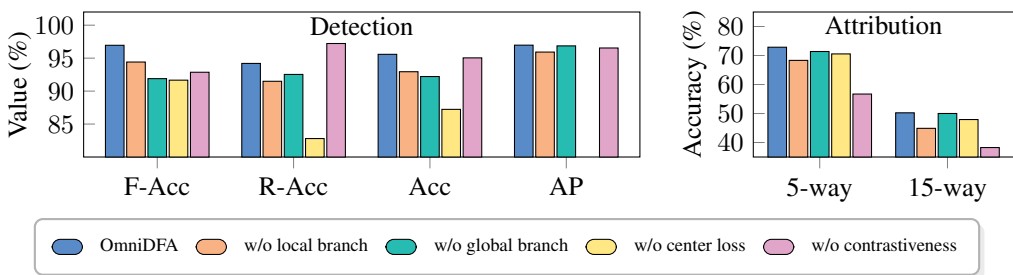

Figure 5: Ablation study on model component.

we found that the higher the degree of Gaussian blur, the more likely an image is to be perceived as synthetic, which is contrary to the common assumption that blurring makes real and synthetic images harder to distinguish. This provides strong empirical support for future model development.

**Component-wise analysis.** To investigate the impact of each component, we perform a series of ablation studies. We analyze the effects of removing the local and global branches, as well as a configuration that excludes the center loss and boundary constraints. Additionally, we train a binary classifier without employing contrastive learning. All models are trained and evaluated under the same experimental setup, and we report the average detection and attribution performance across all three parts of the OmniFake dataset. The results are shown in Figure 5.

The results demonstrate the clear advantage of our dual-path model over the single-branch variants, confirming the superiority of our multi-level feature fusion. Our OmniDFA outperforms a plain binary classifier by capturing finer inter-class distinctions, which consequently enhances fake image detection accuracy. Furthermore, compared to the model trained without boundary constraints, our full framework shows stronger discriminative capability on real images. This underscores the importance of leveraging large-scale, category-rich datasets for training, thereby highlighting a critical advantage of our dataset.

## 5 CONCLUSION

In this paper, we introduce open-set, few-shot source identification, which advances the field of image attribution beyond closed-set recognition and toward practical scenarios. We present Om-niFake, a comprehensively curated dataset containing 1.17 M images from 45 distinct generative models. Our dataset is both sufficiently large and meticulously categorized, enabling detailed investigation into model-specific artifacts. Building upon this foundation, we propose OmniDFA (Omni Detector and Few-shot Attributor), a unified framework based on supervised contrastive learning that jointly performs authenticity detection and few-shot source identification. Experiments show that OmniDFA not only achieves state-of-the-art generalization in synthetic image detection, but also exhibits strong open-set attribution capability with limited reference samples. The results underscore the real-world applicability of our approach, highlighting how explicit modeling of synthesis origins can enhance detector robustness, thereby suggesting a promising path for future research.

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

# A  Necessity of Class-Aware Dataset

Most currently available datasets fail to account for the architectural uniqueness of generative models. For instance, the widely-used GenImage dataset (Zhu et al., 2023) includes three models: Stable Diffusion 1.4, Stable Diffusion 1.5, and Wukong, all of which share identical backbone structures. This structural homogeneity results in remarkably similar feature distributions among their generated images. Such design leads to significant limitations: even a simple CNN classifier trained on images from any one model demonstrates exceptionally strong generalization performance when evaluated on the other two models (Zhu et al., 2023). This architectural redundancy severely compromises the objectivity and comprehensiveness of model evaluation. Additionally, another study (Hong et al., 2025) demonstrates that this generalization capability remains valid for both fine-tuning (Ruiz et al., 2023) and LoRA adaptation (Hu et al., 2022). Therefore, merely making minor modifications to the weights or model architecture is insufficient to enable the model to express fundamentally different representations.

To comprehensively evaluate the generalization ability of deepfake detectors, we argue that models across all categories must be architecturally distinct, rather than just variations of the same backbone. To address these requirements and support our experimental objectives, we introduce OmniFake, a large-scale comprehensive dataset comprising images synthesized by a diverse collection of distinct generative models. This dataset not only enables systematic investigation of detection generalization across architectures, but also facilitates out-of-distribution classification tasks. Our dataset includes an extensive range of state-of-the-art generative models, including autoregressive architectures and unified Multimodal Large Language Models (MLLMs). We believe our dataset will significantly advance research in the fields of both synthetic image detection and attribution.

# B  Related Work

## B.1  Synthetic Datasets

Early datasets for AI-generated image detection primarily relied on Generative Adversarial Networks (Goodfellow et al., 2014). CNNSpot (Wang et al., 2020) proposes a widely-used dataset and establishes an evaluation paradigm where detectors are trained on images from ProGAN (Karras et al., 2018) and LSUN (Yu et al., 2016), then tested across various other models. As diffusion models (Ho et al., 2020) advance in image synthesis, many datasets Epstein et al. (2023b); Ojha et al. (2023a) have incorporated images from these models to evaluate and improve detection methods. GenImage (Zhu et al., 2023) introduces the first million-scale synthetic dataset, paired with real images from ImageNet (Russakovsky et al., 2015). However, these datasets only cover a limited number of generators, restricting the generalization capability of detectors.

Recent studies (Asnani et al., 2023) have begun emphasizing synthetic model diversity in dataset construction. WildFake (Hong et al., 2025) introduces a wild-collected dataset featuring diverse fake images from various generative models, covering a wide range of content and styles. It also reveals that model architecture is the primary factor of bias, outweighing the effects of personalized finetuning (Ruiz et al., 2023) or LoRA adaption (Hu et al., 2022). Community Forensics (Park & Owens, 2025) addresses the diversity limitation by aggregating samples from thousands of generative models. However, these datasets are suboptimal for attribution, as their categories often consist of fine-tuned or adapted versions of the same base model, making it difficult to achieve meaningful distinction. Moreover, their heavy reliance on diffusion models also misaligns with the current paradigm shift toward autoregressive architectures.

## B.2  Detection Methods

Previous research on synthetic image detection has been dedicated to feature extraction from multiple perspectives, such as frequency (Tan et al., 2024b), semantics (Tan et al., 2025), and reconstruction difficulty (Wang et al., 2023a; Luo et al., 2024). Although they have achieved promising results on previous datasets, recent studies (Grommelt et al., 2024; Li et al., 2025a) suggest that data augmentation techniques such as resizing and JPEG compression can significantly degrade the performance of detectors. To extract more robust features, OOC-CLIP (Liu et al., 2024) employs the pre-trained CLIP model as its feature extractor, whereas FakeReasoning (Gao et al., 2025) utilizes

vision-language models which not only effectively incorporate textual prompts but also produce human-interpretable feature descriptions.

Recent studies (Cocchi et al., 2024) have introduced more sophisticated learning objectives to overcome the limitations of binary classifiers. Bi et al. (2023) introduce learning with real images only by analyzing pixel-level distributions. NTF (Liang et al., 2025) employs self-supervised feature mapping to enhance transfer learning, while FSD (Wu et al., 2025b) learns a specialized metric space to distinguish unseen fake images with given samples. Inspired by these advances, we aim to fully explore the potential of metric learning for out-of-distribution deepfake detection.

### B.3 ATTRIBUTION METHODS

Image attribution aims to identify the source model of generated images. Early studies (Frank et al., 2020; Bui et al., 2022) focus on closed-set classification over GANs, where all generators encountered during testing are assumed to be known at training time. Their motivations are based on the observation that models with different architectures exhibit distinct fingerprints. However, the closed-set setting has limited practicality in real-world scenarios, as new generative models are continuously being released. To address this limitation, the open-set rejection task has been introduced (Fang et al., 2023), where a model is required to categorize fake images from unseen generators into an additional rejection class, thereby distinguishing them from known categories. For instance, DNA-Det (Yang et al., 2022) captures globally consistent architectural traces through patch-based contrastive learning. CPL (Sun et al., 2023) introduces a global-local voting module to evaluates attribution performance on various GAN-generated face images. DE-FAKE (Sha et al., 2023) employs the CLIP model to attribute fake images created by text-to image diffusion models.

Nevertheless, most methods still struggle to fully exploit fine-grained cross-modal fingerprints and show limited generalization capability in zero-shot scenarios when confronted with novel generators. We argue that the open-set rejection formulation remains distant from real-world applications, as the performance of such methods tends to degrade when insufficient data is available for continuously emerging generative models. Therefore, we propose an open-set few-shot paradigm to mitigate data scarcity issues and evaluate model performance from detecting forgery clues from unseen, novel generators. This necessitates that the model rapidly learns to identify salient features from limited samples, while also distinguishing them from features characteristic of other models.

## C COLLECTION OF FAKE SAMPLES

### C.1 FAKE IMAGE COMPOSITION

Our dataset comprises synthetic images collected through three primary sources to ensure diversity and comprehensiveness. First, we incorporate images from established open-source datasets and benchmarks, including GenImage (Zhu et al., 2023), WildFake (Hong et al., 2025), and MPBench (Lu et al., 2023), covering a wide range of generative models such as GANs, diffusion models, and VAEs. This subset consists of images generated by 19 distinct generators, providing a solid foundation of varied synthetic content. Second, we augment our dataset with community-shared collections from platforms like Hugging Face (HuggingFace, 2016), integrating synthetic images from 6 closed-source models and 2 popular community models. With the rapid advancement of generative models, many previously state-of-the-art architectures have become outdated in terms of both design and performance. To ensure our study reflects the latest progress in the field, we sourced 18 cutting-edge open-source models from Hugging Face or the official repositories. We generate synthetic images using these models to ensure the coverage of diverse architectures such as flow-matching models, autoregressive models and unified MLLMs. We summarize the key details of these models in Table 5.

Our OmniFake contains 1.17 M synthetic images generated by 45 distinct generative models, spanning a broad spectrum of architectures. Our synthetic categories include models from the same family, but we rigorously ensure that they are not derived from the same backbone. To ensure both diversity and detectability, each synthetic category includes at least 20 K training samples generated from a wide distribution of text prompts. For comprehensive evaluation, we provide a separate test set of 90 K synthetic images (2 K per category). Our dataset features the most structurally diverse

Table 5: Collections of generators for fake images in our OmniFake dataset.

| Generator | Training | Test | Source | Link |
|---|---|---|---|---|
| ADM | 30k | 2k | | |
| GLIDE | 30k | 2k | | |
| Midjourney V5 | 30k | 2k | GenImage | https://github.com/GenImage-Dataset/GenImage |
| Stable Diffusion V1.5 | 30k | 2k | | |
| VQDM | 30k | 2k | | |
| DALLE2 | 30k | 2k | | |
| StyleGAN3 | 30k | 2k | | |
| DF-GAN | 30k | 2k | | |
| GALIP | 30k | 2k | | |
| GigaGAN | 25k | 2k | | |
| DDIM | 30k | 2k | WildFake | https://github.com/hy-zpg/AIGC-Image-Detection-Dataset |
| DDPM | 30k | 2k | | |
| Imagen | 30k | 2k | | |
| Midjourney V4 | 30k | 2k | | |
| SDXL | 30k | 2k | | |
| VQVAE | 30k | 2k | | |
| Muse | 30k | 2k | | |
| IF | 30k | 2k | Fake Image Dataset | https://huggingface.co/datasets/InfImagine/FakeImageDataset |
| Cogview2 | 20k | 2k | | |
| FLUX-dev | 30k | 2k | | https://huggingface.co/datasets/lehduong/flux_generated |
| GPT4-o | 30k | 2k | | https://huggingface.co/datasets/yufan/GPT4O_Image_T2I |
| DALLE3 | 30k | 2k | | https://huggingface.co/datasets/OpenDatasets/dalle-3-dataset |
| Phoenix | 30k | 2k | | https://huggingface.co/datasets/bigdata-pw/leonardo/ |
| PixArt-Alpha | 27k | 2k | Hugging Face | https://huggingface.co/datasets/PixArt-alpha/PixArt-Eval30K |
| Playground V2.5 | 30k | 2k | | https://huggingface.co/datasets/bigdata-pw/playground |
| Ideogram | 30k | 2k | | https://huggingface.co/datasets/terminusresearch/ideogram-75k |
| Midjourney V6 | 30k | 2k | | https://huggingface.co/datasets/terminusresearch/midjourney-v6-520k-raw |
| DiT-XL/2 | 23k | 2k | | https://github.com/facebookresearch/DiT |
| Janus-Pro | 24k | 2k | | https://github.com/deepseek-ai/Janus |
| BAGEL | 23k | 2k | | https://github.com/ByteDance-Seed/Bagel |
| OmniGen | 23k | 2k | | https://huggingface.co/Shitao/OmniGen-v1 |
| SD3-Medium | 23k | 2k | | https://huggingface.co/stabilityai/stable-diffusion-3-medium |
| Hunyuan-DiT | 23k | 2k | | https://huggingface.co/Tencent-Hunyuan/HunyuanDiT-v1.2-Diffusers |
| Show-o | 23k | 2k | | https://github.com/showlab/Show-o |
| LUMINA-Image 2.0 | 23k | 2k | | https://huggingface.co/Alpha-VLLM/Lumina-Image-2.0 |
| SANA V1.5 | 20k | 2k | Self-synthesized | https://huggingface.co/Efficient-Large-Model/SANA1.5_4.8B_1024px_diffusers |
| CogView4 | 20k | 2k | | https://huggingface.co/zai-org/CogView4-6B |
| OmniGen2 | 20k | 2k | | https://github.com/VectorSpaceLab/OmniGen2 |
| HiDream-I1-Dev | 20k | 2k | | https://github.com/HiDream-ai/HiDream-I1 |
| Infinity | 20k | 2k | | https://github.com/FoundationVision/Infinity |
| Llama-Gen | 20k | 2k | | https://github.com/FoundationVision/LlamaGen |
| UniWorld-V1 | 20k | 2k | | https://github.com/PKU-YuanGroup/UniWorld-V1 |
| BLIP3-o | 20k | 2k | | https://github.com/JiuhaiChen/BLIP3o |
| BRIA3.2 | 20k | 2k | | https://huggingface.co/briaai/BRIA-3.2 |
| Ovis-U1 | 20k | 2k | | https://github.com/AIDC-AI/Ovis-U1 |

generator categories, along with the latest, most extensive, and top-tier generators. OmniFake contains images with a minimum resolution of $200 \times 200$ pixels, while most are $512 \times 512$ or higher. This comprehensive collection provides a robust foundation for investigating the impact of model architectures on generalization.

## C.2 CUSTOM DATA SYNTHESIS

To ensure the diversity and richness of prompts in our self-synthetic categories, we sample text descriptions from multiple text-image datasets or benchmarks, covering different granularities of textual inputs. Specifically, we adopt BLIP-3o-60k (Chen et al., 2025a) for fine-grained, highly detailed captions, CC12M (Changpinyo et al., 2021) for coarse-grained and concise prompts, and Laion-COCO (Schuhmann et al., 2021) for in-the-wild descriptions. To maintain representativeness and diversity in prompt selection, we balance the sampling ratio among these sources at $2 : 5 : 3$, effectively integrating their respective characteristics. Based on these curated prompt lists, we leverage 17 distinct generative models to synthesize images, maximizing variety in the generated outputs. To ensure optimal image generation quality, we adhere to the official recommended resolutions. When multiple resolutions are provided, we randomly select among them to maintain diversity in the output image dimensions. For DiT-XL/2, which only accept class labels as conditional input, we randomly sample from the categories for image generation. Although this approach may introduce certain domain biases, we argue that the pronounced artifacts inherent to the synthetic generators could potentially overshadow such biases. Ultimately, our multi-source, multi-granularity prompt sampling strategy ensures broad coverage and high quality in the synthetic dataset.

Table 7: Split of our OmniFake dataset.

| Split Part | Test Generators |
|---|---|
| Part I | Hunyuan-DiT, Imagen, SANA V1.5, DF-GAN, Janus-Pro, DDPM, Midjourney V5, OmniGen2, FLUX-dev, BRIA3.2, Ovis-U1, Cogview2, VQVAE, Phoenix, DIT-XL/2 |
| Part II | OmniGen, LUMINA-Image 2.0, Show-o, SD3-Medium, Midjourney V6, GALIP, LlamaGen, Ideogram, Infinity, Muse, StyleGAN3, ADM, IF, GigaGAN, VQDM |
| Part III | SDXL, Playground V2.5, UniWorld-V1, BLIP3-o, Midjourney V4, CogView4, PixArt-Alpha, DALLE2, DDIM, GLIDE, GPT4-o, HiDream-I1-Dev, BAGEL, DALLE3, Stable Diffusion V1.5 |

## D  COLLECTION OF REAL SAMPLES

Prior image forensics benchmarks (Wang et al., 2020; Zhu et al., 2023) often rely on a single source dataset to represent authentic images. However, this practice can introduce significant real biases by limiting the diversity and representativeness of the authentic class. To mitigate this limitation and establish a more robust and generalizable benchmark, we follow Hong et al. (2025) and curate our authentic image collection by strategically sampling from a diverse set of 10 publicly available datasets, spanning multiple domains and collection paradigms. An overview of the source datasets is provided in Table 6.

Our authentic image dataset is constructed from two complementary categories of publicly available sources to ensure both broad coverage and high quality. Large-scale raw datasets such as LAION (Schuhmann et al., 2021), WuKong (Gu et al., 2022), and CC12M (Changpinyo et al., 2021) provide essential in-the-wild diversity by capturing the unfiltered heterogeneity of real-world internet imagery at scale, which is crucial for evaluating forensic models under realistic conditions. These are balanced with carefully constructed datasets including ImageNet (Russakovsky et al., 2015), MSCOCO (Lin et al., 2014) and and several other representative collections, which offer domain-specific focus through their manually curated collections of high-quality images representing well-composed photographic contexts.

Table 6: Collection of datasets for real images in our OmniFake dataset.

| Real Image Dataset | Training | Test |
|---|---|---|
| Laion-5B | 251k | 20k |
| Wukong | 242k | 20k |
| ImageNet-1k | 174k | 15k |
| CC12M | 160k | 15k |
| MSCOCO | 113k | 10k |
| FFHQ | 68k | 2k |
| CelebA-HQ | 28k | 2k |
| LSUN-church | 80k | 2k |
| IMD2020 | 33k | 2k |
| FODB | 21k | 2k |

To conserve resources and avoid dataset conflicts, we strategically sourced our LAION and Wukong samples from the training set of WildFake, while drawing ImageNet samples from the GenImage training collection.

We allocate varying proportions based on dataset size and our requirements, obtaining 1.17 million authentic images for our training set. We also select 90 K real images matching the size of the fake images to serve as our test set. By integrating these different components to dataset construction, we aim to achieve a balanced representation of both uncontrolled web content and professionally captured images.

## E  EXPERIMENTS SETTINGS

### E.1  SPLIT OF OMNIFAKE

To ensure a fair evaluation in our zero-shot task, we employ a 3-fold cross-validation strategy on the OmniFake dataset. The dataset is randomly divided into three mutually exclusive and balanced parts, as shown in Table 7. In each validation round, two folds are used for training, while the remaining fold serves as the held-out test set. This approach not only maximizes the utilization of limited data but also guarantees that model performance is evaluated across diverse and representative subsets.

## E.2 IMPLEMENTATION DETAILS

To guarantee the reproducibility of our experiments, we present comprehensive training details for our model in this work, which we summarize in Table 8 and Table 9. In Table 8, p denotes the probability of applying the corresponding transformation. It should be noted that in RandAugment (Cubuk et al., 2020), we deliberately exclude shear and translate transformations to ensure that our local feature extractor does not capture artifacts from padding regions beyond image boundaries. The data augmentation is only employed during the training phase. In our cross-dataset validation experiments, we adopted the following hyperparameters: $40$ training epochs and a $\lambda$ value of $5e-3$, which are carefully selected to enhance the robustness of our results. Our implementation is carried out using PyTorch library, with all experiments executed on a cluster of 8 NVIDIA A100 GPUs.

Table 8: Data augmentation configurations during training.

| Augmentation | Settings |
| --- | --- |
| random JPEG | p=0.5, quality=(75, 95) |
| resize | scale=(0.5, 2.0) |
| horizontal flip | p=0.5 |
| RandAugment | p=0.5, magnitude=9, layers=2 |
| gaussian blur | p=0.5, sigma=(0.1, 2.0) |
| normalize | [0,1] |

Table 9: Training hyperparameters for our experiments.

| Hyperparameter | Value |
| --- | --- |
| model | ConvNeXt-Small |
| MLP hidden dims | 512 |
| MLP out dims | 128 |
| input resolution | $3\times224\times224$ |
| batch size (fake, per GPU) | 128 |
| batch size (real, per GPU) | 16 |
| total epochs | 20 |
| warmup epochs | 2 |
| optimizer | AdamW |
| scheduler | CosineAnnealing |
| learning rate | 2e-5 |
| min learning rate | 0 |
| weight decay | 1e-2 |
| precision | bfloat |
| world size | 8 |
| $\tau$ | 0.07 |
| $\lambda$ | 0.01 |
| $\beta$ | 0.99 |