# OpenReview forum: "OmniDFA: A Unified Framework for Open Set Synthesis Image Detection and Few-Shot Attribution"
_ICLR.cc/2026/Conference — ICLR 2026 Conference Withdrawn Submission_

### Official Review · Reviewer_eKfg · 2025-10-30

**Soundness:** 1
**Presentation:** 2
**Contribution:** 1
**Rating:** 2
**Confidence:** 5

**Summary:**

This paper proposes an AIGC detection method from the few-shot perspective. It aims to provide additional category information from the generator’s viewpoint and uses supervised contrastive learning for optimization. Furthermore, based on the reorganization of existing datasets, the OmniFake dataset is constructed. Experiments on this dataset demonstrate certain advantages over existing works.

**Strengths:**

- It reorganizes existing datasets to form the OmniFake dataset.
- It uses contrastive learning to characterize model-specific artifacts and integrates low- and high-level features to build the OmniDFA detector.
- It demonstrates certain effectiveness on datasets such as OmniFake and GenImage.

**Weaknesses:**

- The claim in Contribution 1 that "few-shot identification as a previously unexplored task" lacks sufficient literature review. Exploring AIGC detection from the few-shot (or open-set) perspective is not a first attempt. For example, the work "Raising the bar of AI-generated image detection with CLIP" was presented at CVPR'24.

- The term "Distinct" in Table 1 is highly imprecise. What defines a "distinct architecture"? Does a difference in model parameter count make an architecture "distinct"? Or do architectures with similarities (e.g., ResNet-18 and ResNet-50) belong to the "same architecture"? Why does the proposed OmniFake dataset cover 45 "distinct" architectures while other methods are not considered to involve "distinct" ones? Additionally, the fake images in the OmniFake dataset are sourced from existing datasets such as GenImage and WildFake (as stated in Line 150), yet the authors classify GenImage and WildFake as non-"Distinct" in Table 1. Finally, since the data in OmniFake is merely a reorganization of existing datasets, it cannot be regarded as a new dataset. Thus, the contribution of the constructed OmniFake dataset is overstated, and the comparison with existing datasets is inaccurate.

- Insufficient technical innovation:
(1) Extracting low- and high-level features is a commonly used approach, but the paper fails to explain the necessity of adopting this method and its connection to multiple categories. (2) There is no reasonable explanation for why images generated by different architectures possess distinct forensic traces and thus are opposed to each other in the contrastive loss. What if some architectures generate identical traces? The use of contrastive loss lacks a valid rationale, and the paper does not conduct an in-depth exploration of the correlation between the fake image space and its original generation architecture. (3) Moreover, Sections 3.2 and 3.3 simply adopt existing contrastive loss and center loss without making effective improvements tailored to the AIGC detection task.

- Since the OmniFake dataset explicitly includes GenImage during its construction, why is the detection of GenImage classified as zero-shot in Table 4? Additionally, in Table 4, the comparative algorithms (e.g., CNNSpot and UnivFD) and the proposed method are trained on different datasets; comparing them together here is meaningless.

- From the results in Fig. 5, removing the local branch has no impact on the AP metric. Given this, what is the motivation for using the local branch as described in Section 3?

- In Fig. 1, the borders of "in set" and "real" are blue and green respectively, which are difficult to distinguish.

**Questions:**

See Weaknesses.

---

### Official Review · Reviewer_NR7M · 2025-11-01

**Soundness:** 3
**Presentation:** 3
**Contribution:** 2
**Rating:** 6
**Confidence:** 5

**Summary:**

This paper introduces OmniDFA, a unified framework for both AI-generated image (AIGI) authenticity detection and open-set few-shot source attribution. The authors further provide OmniFake, a large-scale and category-aware synthetic image dataset covering 45 generative models across various architectures. OmniDFA employs a dual-path feature extractor to capture local and global visual signals, along with contrastive learning and a sphere center loss to improve discrimination. Experiments demonstrate superior performance over existing approaches on detection and attribution tasks, particularly under zero-shot and few-shot settings.

**Strengths:**

- The OmniFake dataset is a meaningful contribution that fills a gap in large-scale, category-aware synthetic image benchmarks. It includes a diverse set of 45 generative models, enabling more rigorous investigation into model-specific attribution. This dataset will likely benefit future research in both AIGI detection and source tracing.
- The integration of authenticity detection with open-set, few-shot attribution in a single framework is practically impactful. This unified approach aligns well with real-world forensic needs where both detection and model tracing are required. It reduces pipeline redundancy and improves applicability compared to prior work focusing on one task only.
- The model consistently outperforms competitive baselines across multiple benchmarks, including zero-shot settings on GenImage and Chameleon. The robustness analysis further highlights resilience to image compression and blur within reasonable ranges. These results suggest that the framework generalizes well and holds practical promise for deployment in unconstrained environments.

**Weaknesses:**

- The dual-path feature design for capturing low-level and high-level features is conceptually incremental, as similar multi-scale hybrid modeling has been employed in prior forensics and vision models. The paper does not sufficiently justify why this design is uniquely effective for AIGI detection and attribution. A deeper comparative ablation against more advanced multi-scale architectures (e.g., Swin variants, multi-level ViT) would strengthen the novelty claim
- The use of contrastive learning combined with sphere center loss seems effective but lacks theoretical explanation tailored to AIGI properties. Center loss is not new and has been widely used in face recognition and metric learning, thus requiring clearer justification for its necessity and superiority here. Additional experiments comparing with alternative losses (e.g., ArcFace, triplet loss, prototype-based loss) would improve the contribution.
- Few-shot and open-set attribution is a key claim, yet comparisons to state-of-the-art few-shot or OOD/meta-learning methods are missing. The evaluation lacks analysis on how performance scales with the number of novel classes and number of shots beyond the limited scenarios shown. Including baselines such as Prototypical Networks, Matching Networks, or OOD detection methods would make the attribution claim more convincing.

**Questions:**

- Does the dual-path design offer clear advantages over more principled multi-scale feature extractors such as hierarchical ViT or feature pyramids? Can you provide comparisons?
- Why was center loss chosen instead of more advanced metric-learning losses (e.g., ArcFace, triplet, prototypical loss)? What specific property of AIGI data makes center loss preferable?
- Can you provide results on attribution performance as the number of unseen generators increases or number of shots decreases?
- How robust is the attribution component of OmniDFA to real-world perturbations beyond JPEG compression and Gaussian blur？
- The baselines used in Table 3 and Table 4 are not aligned, which makes it difficult to directly compare the results across settings. Could the authors clarify the criteria for selecting different baseline methods for these experiments?

---

### Official Review · Reviewer_sa51 · 2025-11-01

**Soundness:** 2
**Presentation:** 2
**Contribution:** 1
**Rating:** 2
**Confidence:** 5

**Summary:**

This paper introduces OmniDFA, a unified framework for open-set AI-generated image detection and few-shot source attribution, targeting the practical challenge of identifying synthetic content from previously unseen generative models with limited reference samples. To support this, the authors construct OmniFake, a large-scale, class-aware dataset comprising images from distinct generative models and a matching set of real images. OmniDFA employs a dual-path architecture with supervised contrastive learning and center loss to achieve strong generalization for both detection and attribution tasks. Extensive experiments verify the efficiency of the proposed method.

**Strengths:**

1. This paper introduces the open-set few-shot source attribution task, exploring the attribution of unknown models under few-shot conditions, framing a practical and scalable challenge that better reflects real-world needs compared to closed-set or naive open-set rejection.

2. The authors introduce OmniFake, a comprehensive and diverse synthetic image dataset capturing structural heterogeneity across 45 generator categories, supporting the real-world attribution scenarios.

3. The paper employs a dual-stream detection model, which enhances the extraction of image source attribution features by preserving global semantic content and maintaining local details. Ablation studies validate the effectiveness of this module.

**Weaknesses:**

1. The literature review in this paper is insufficient, lacking citations of recent work on open-set source attribution methods, e.g., POSE (Yang et al. CVPR 2023). Furthermore, the experimental comparisons omit state-of-the-art open-set zero-shot source attribution approaches.

2. The experimental comparison settings are unfair. For example, in Table 4, other methods are trained on a single type of generated data, whereas the proposed method is trained on data from multiple models. Therefore, the performance improvement does not convincingly demonstrate the effectiveness of the proposed approach.

3. The authors have an incorrect definition of the source attribution task. In practical scenarios, the task should include tracing models that have been fine-tuned from the same base model but with different LoRA adaptations. However, the dataset constructed in this paper does not encompass such scenarios, resulting in an incomplete formulation of the source attribution problem.

**Questions:**

1. Can the proposed method perform source attribution when dealing with the same model architecture but different LoRA adaptations, and to what extent would the attribution results be affected by factors such as random seeds?

2. In the zero-shot binary classification task, if other methods were also trained on multi-model data, how much would their generalization performance improve, and how would this compare with the proposed approach?

---

### Note · Authors · 2025-11-13

I have read and agree with the venue's withdrawal policy on behalf of myself and my co-authors.